# VIDEO-INFINITY:
# DISTRIBUTED LONG VIDEO GENERATION

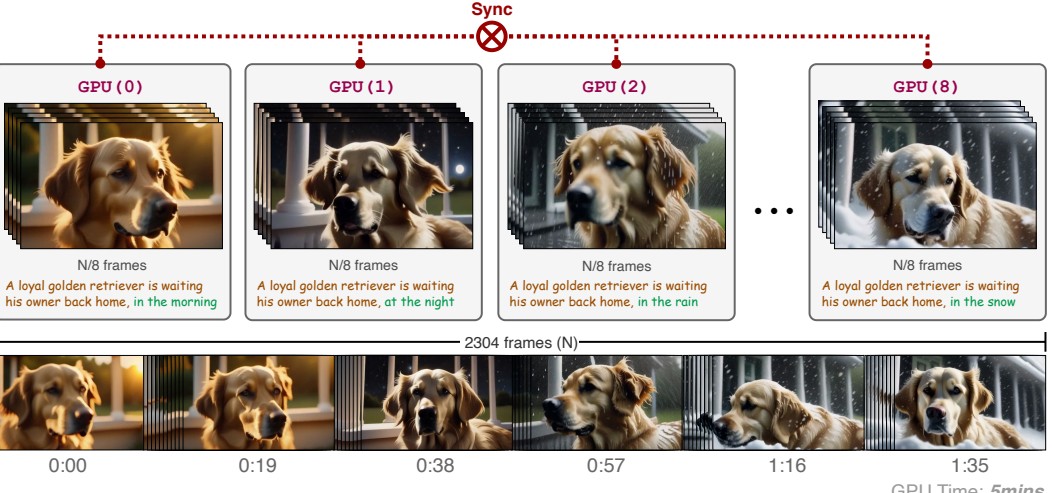

Figure 1: Multiple GPUs parallelly generate a complete video, producing 2300 frames in 5 minutes.

## ABSTRACT

Diffusion models have recently achieved remarkable results for video genera-
tion. Despite the encouraging performances, the generated videos are typically
constrained to a small number of frames, resulting in clips lasting merely a few
seconds. The primary challenges in producing longer videos include the substantial
memory requirements and the extended processing time required on a single GPU.
A straightforward solution would be to split the workload across multiple GPUs,
which, however, leads to two issues: (1) ensuring all GPUs communicate effec-
tively to share timing and context information, and (2) modifying existing video
diffusion models, which are usually trained on short sequences, to create longer
videos without additional training. To tackle these, in this paper, we introduce
Video-Infinity, a distributed inference pipeline that enables parallel processing
across multiple GPUs for long-form video generation. Specifically, we propose two
coherent mechanisms: *Clip parallelism* and *Dual-scope attention*. Clip parallelism
optimizes the gathering and sharing of context information across GPUs which
minimizes communication overhead, while Dual-scope attention modulates the
temporal self-attention to balance local and global contexts efficiently across the
devices. Together, the two mechanisms join forces to distribute the workload and
enable the fast generation of long videos. Under an $8 \times$ Nvidia 6000 Ada GPU
(48G) setup, our method generates videos up to 2,300 frames in 5 minutes.

## 1 INTRODUCTION

A long-standing pursuit of human beings is to replicate the dynamic world we live in, in the digital
system. Traditionally dominated by physics and graphics, this effort has recently been enhanced by
the emergence of data-driven generative models (Rombach et al., 2022; Ho et al., 2022b; Harvey
et al., 2022; Ho et al., 2022a), which can create highly realistic images and videos indistinguishable

from reality. However, these models typically produce very short video segments, with most limited to 16-24 frames (Guo et al., 2023; Chen et al., 2023a; 2024). Some models extend to 60 or 120 frames (Zhaoyang et al., 2024; hpcaitech, 2024), but compromise heavily on resolution and visual quality.

Generating long videos poses substantial challenges, primarily due to the extensive resource demands for model training and inference. Current models, constrained by available resources, are often trained on brief clips, making it difficult to sustain quality over longer sequences. Moreover, generating a minute-long video in one go can overwhelm GPU memory, making the task seem elusive.

Existing solutions, including autoregressive, hierarchical, and short-to-long methods, offer partial remedies but have significant limitations. Autoregressive methods (Henschel et al., 2024; Yin et al., 2023) produce frames sequentially, dependent on preceding ones. Hierarchical methods (Chen et al., 2023b; Yin et al., 2023; Zhou et al., 2024) create keyframes first, then fill in transitional frames. Furthermore, some approaches treat a long video as multiple overlapping short video clips (Qiu et al., 2023; Wang et al., 2023a). These methods are not end-to-end; they often miss global continuity, require extensive computation, especially in regions of overlap, and struggle with consistency across different segments.

To bridge these gaps, we introduce a novel framework for distributed long video generation, termed Video-Infinity. On the high level, it works in a divide-and-conquer principle. It breaks down the task of long video generation into smaller, manageable segments. These segments are distributed across multiple GPUs, allowing for parallel processing. All clients should work collaboratively to ensure the final video is coherent in semantics.

This setup, while straightforward, faces two principal challenges: ensuring effective communication among all GPUs to share contextual information, and adapting existing models—typically trained on shorter sequences—to generate longer videos without requiring additional training.

To overcome these challenges, we introduce two synergistic mechanisms: *Clip parallelism* and *Dual-scope attention*. Clip parallelism enables efficient collaboration among multiple GPUs by splitting contextual information into three parts. It uses an interleaved communication strategy to complete the sharing in three steps. Building on the capabilities of Clip parallelism, Dual-scope attention meticulously adjusts the temporal self-attention mechanisms to achieve an optimal balance between local and global contexts across devices. This balance allows a model trained on short clips to be extended to long video generation with overall coherence.

Even more exciting, by leveraging both strategies, Video-Infinity reduces memory overhead from a quadratic to a linear scale. With the power of multiple device parallelism and sufficient VRAM, our system can generate videos of any, potentially even infinite length.

As a result, our method significantly extends the maximum length of videos that can be generated and accelerates the speed of long video generation. Specifficly, on an 8 × Nvidia 6000 Ada (48G) setup, our method manages to generate videos up to 2300 frames in just 5 minutes. Our contributions are summarized as follows: (1) We are the first to address long video generation using distributed parallel computation, enhancing scalability and reducing generation times. (2) We introduce two interconnected mechanisms: Clip parallelism, which optimizes context information sharing across GPUs, and Dual-scope attention, which adjusts temporal self-attention to ensure video coherence across devices. (3) Our experiments show that our approach is over 100 times faster than the recent work Streaming T2V (Henschel et al., 2024) and 10 times faster than the concurrent work FIFO-Diffusion (Kim et al., 2024) when generating ultra-long videos.

## 2 RELATED WORKS

### 2.1 DIFFUSION MODELS

Diffusion models have gained significant attention in recent years due to their impressive ability to generate high-quality media. Originally introduced for image synthesis, models like Denoising Diffusion Probabilistic Models (DDPM) (Ho et al., 2020) and Latent Diffusion Models (LDM) (Rombach et al., 2022) have demonstrated state-of-the-art performance in image generation. These models progressively denoise a Gaussian noise distribution by learning a sequence of reverse transformations.

Beyond images (Ho et al., 2020; Rombach et al., 2022), diffusion models have also shown promise in audio (Kong et al., 2020; Yang et al., 2023; Liu et al., 2023) and 3D generation (Luo & Hu, 2021; Poole et al., 2022). Adaptations of diffusion models for video generation incorporate temporal modules to capture the sequential nature of video frames. For instance, Video Diffusion Models (VDM) (Ho et al., 2022b) and Flexible Diffusion Model (FDM) (Harvey et al., 2022) effectively extend diffusion frameworks to video data, overcoming challenges like temporal consistency and quality degradation. More recent models such as AnimateDiff (Guo et al., 2023), ModelScope (Wang et al., 2023b), and VideoCrafter (Chen et al., 2023a; 2024) can now produce video clips with better dynamics and improved visual quality.

## 2.2 Techniques for long video generation

Streaming T2V (Henschel et al., 2024) introduces a method that relies on a conditional attention module to ensure smooth transitions between video segments and a scene-preserving mechanism for content consistency. However, this method requires training and is not end-to-end, posing limitations on its practicality. FreeNoise (Qiu et al., 2023) utilizes rescheduled noise sequences and window-based temporal attention to improve video continuity. Despite these innovations, the rescheduled noise contributes to limited dynamics in the generated videos, and the overlapping attention windows introduce additional computational overhead. The concurrent work, FIFO-Diffusion (Kim et al., 2024) employs a sliding pipeline to achieve longer video generation, also leveraging multiple GPUs, but falls short in efficiency (see Table 1). Another concurrent work, FreeLong (Lu et al., 2024), blends global low-frequency video features with local high-frequency details to maintain consistency and fidelity in long video generation. However, it is still limited to single-GPU generation.

## 2.3 Distributed diffusion

Recently, various distributed parallel methods have been applied to diffusion models to reduce the latency of each denoising step in diffusion models. ParaDiGMS (Shih et al., 2024) utilizes step-based parallelism, where each denoising step is executed on a different GPU device in parallel. However, this approach tends to waste much computation. Another method, DistriFusion (Li et al., 2024), divides images into patches, allowing different patches to be denoised on separate GPUs. This approach ensures synchronization among patches and achieves minimal computational waste. However, it is designed specifically for image diffusion and requires significant communication overhead and specialized hardware support to achieve low latency.

## 3 Preliminaries

**Diffusion Models in Video Generation**

The process of generating videos using diffusion models involves progressively denoising the latent representation, denoted as $x_t$, where $t$ ranges from 0 to $T$. The initial noisy video latent is represented by a random noise tensor $x_T$. With each denoising step, $x_t$ is updated to a clearer latent $x_{t-1}$. This iterative process continues until $x_T$ is denoised to $x_0$, which is then fed into a decoder to generate the final video. The key aspect of updating $x_t$ to $x_{t-1}$ is computing the noisy prediction $\epsilon_t$, given by: $\epsilon_t = \mathcal{E}_\theta(x_t)$. where $\mathcal{E}_\theta$ represents the diffusion model.

The diffusion model $\mathcal{E}_\theta$ can be implemented using various architectures, such as U-Net (Ronneberger et al., 2015; Ho et al., 2022b; Harvey et al., 2022; Guo et al., 2023; Chen et al., 2023a) or DiT (Peebles & Xie, 2023; hpcaitech, 2024; Zhaoyang et al., 2024). These diffusion models are generally composed of several similar layers. More specifically, the initial random noise tensor is written as $x_T \in \mathbb{R}^{F \times H \times W \times C}$, where $F$ represents the number of frames, $H$ and $W$ denote the height and width of each frame, respectively, and $C$ is the number of channels.

The latent tensor $v$ in each layer generally maintains a consistent shape, $v \in \mathbb{R}^{F \times H' \times W' \times C'}$, where $F$ remains constant across layers. The dimensions $H'$, $W'$, and $C'$ can vary due to the down-sampling and up-sampling operations of the U-Net architecture.

These layers in the diffusion model $\mathcal{E}_\theta$ are usually composed of two main types of modules: spatial and temporal. The spatial modules receive slices of the latent $v$ shaped $v \in \mathbb{R}^{H' \times W' \times C'}$ (a single

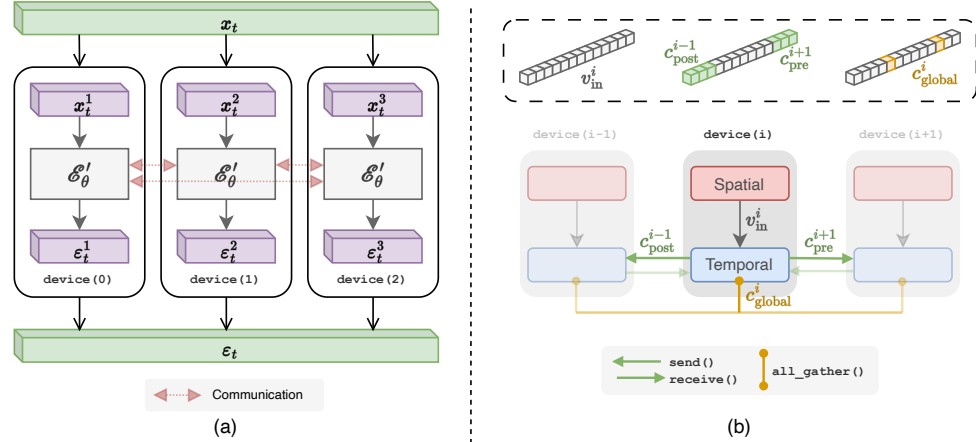

Figure 2: (a) **Pipeline of Video-Infinity**: The latent tensor is split into clips and distributed to different devices. The diffusion model predicts noise in parallel with communication, and the noises are concatenated to produce the final output. (b) **Illustration of Clip parallelism**: In each layer of the video diffusion module, spatial modules operate independently, whereas temporal modules synchronize context elements $c_{\text{pre}}^i$, $c_{\text{post}}^i$, and $c_{\text{global}}^i$. Peer-to-peer and collaborative communications are employed.

frame), representing tokens for each video frame in the latent space. They independently process spatial features within each frame. The temporal modules receive elongated strips of the latent tensor $v$ shaped $v \in \mathbb{R}^{F \times C'}$, representing tokens containing temporal information across frames at specific spatial locations. They capture temporal dependencies between frames at each location.

## 4 DISTRIBUTED LONG VIDEO GENERATION

At the core of our pipeline, Video-Infinity segments the video latent into chunks, which are then distributed across multiple devices. An overview of our method is shown in Figure 3, where we divide the video latent along the temporal dimension. Such partitioning allows for parallel denoising on different devices, each handling non-overlapping frames. To facilitate this, we propose Clip parallelism, detailed in Section 4.1, a mechanism that efficiently synchronizes temporal information across devices. Additionally, we incorporate Dual-scope attention in Section 4.2, which modulates temporal attention to ensure training-free long video coherence and reduces the cost of context synchronization.

Formally, Video-Infinity splits the noisy latent $x_T \in \mathbb{R}^{F \times H \times W \times C}$ into $N$ sub-latent clips $x_T^i \in \mathbb{R}^{F_{\text{clip}} \times H \times W \times C}$, where $i \in [1, N]$, $F_{\text{clip}} = F/N$ represents the number of frames in each clip, and $N$ represents the total number of clips. This structured segmentation facilitates an even load distribution across $N$ devices. Additionally, the spatial modules of video diffusion models operate independently across frames, which eliminates the need for inter-device communication and maintains consistency in the outputs across different devices.

### 4.1 CLIP PARALLELISM FOR VIDEO DIFFUSION

To ensure coherence among clips distributed on different devices, we propose Clip parallelism, shown in Figure 3. It parallelizes the temporal layers for video diffusion models and enables efficient inter-device communication.

**Parallelized temporal modules.** We first design a general structure for any specific type of temporal module, such as convolutional or attention-based modules. It aims to produce the identity result with the original module while minimizing communication costs.

In the standard diffusion model, a temporal module aggregates features across frames, which could be simplified as

$$v_{\text{out}} = \text{temporal}\left(v_{\text{in}}\right), \tag{1}$$

where $v_{\text{in}} \in \mathbb{R}^{F \times * \times C'}$ is the input feature of this temporal layer. However, Video-Infinity distributes input feature tensors $v_{\text{in}}$ across multiple devices, dividing them into several clips $v_{\text{in}}^{i} \in \mathbb{R}^{F_{\text{clip}} \times * \times C'}$, each placed on `device(i)`. To facilitate distributed inference, we introduce the context input $c^i$, which is synchronized from other devices (the specific context $c_i$ for different type temporal modules will be elaborated in the following paragraph ). This enables that the distributed output to remain consistent with the non-distributed result while maintaining the original structure of the temporal modules. The updated temporal operation is defined as:

$$v_{\text{out}}^{i} = \text{temporal}_{\text{Parallel}}\left(v_{\text{in}}^{i}, c^i\right) \tag{2}$$

The output for each device, $v_{\text{out}}^{i}$, reflects localized computations augmented by these contextual inputs. The complete output of the layer, $v_{\text{out}}$, is obtained by concatenating the outputs from all devices:

$$v_{\text{out}} = \text{Concat}\left(\left\{v_{\text{out}}^{i} | i \in [1, N]\right\}\right) \tag{3}$$

This concatenation provides a holistic view of the processed features, yielding the same output as the non-distributed temporal module.

Each $c^i$ includes temporal information from the preceding `device(i-1)` via $c_{\text{pre}}^{i}$, and from the succeeding `device(i+1)` via $c_{\text{post}}^{i}$. Furthermore, $c_{\text{global}}$ is a selective aggregate of inputs from all devices, optimizing global information coherence and reducing overhead.

**Three-round context communication.** Based on the parallelized temporal modules, we present a well-crafted communication mechanism for our Clip parallelism, carefully planning context propagation across devices to transmit all necessary context at the lowest possible communication cost by: 1) minimizing unnecessary `all_gather()` operations, and 2) enabling parallel point-to-point communication to further improve communication efficiency.

To achieve this, we first refine the context $c^i$ in the parallelized temporal modules into $c_{\text{global}}, c_{\text{pre}}^{i}$ and $c_{\text{post}}^{i}$. The shared context, $c_{\text{global}}$, is provided to all devices, while $c_{\text{pre}}^{i}$ and $c_{\text{post}}^{i}$ are obtained from preceding and subsequent devices, respectively. The specific contents of these context components will be elaborated in the following sections.

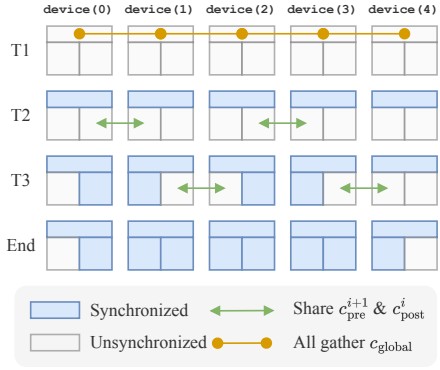

Figure 3: Tree different stages in the communication process of Clip parallelism

We then introduce a three-stage synchronization process, with each stage addressing a specific part of the context. In the first stage, T1, each `device(i)` broadcasts its global context $c_{\text{global}}^{i}$ with all other devices through an `all_gather()` operation. The subsequent stages, T2 and T3 focus on exchanging neighboring contexts. Due to connection limits[1], we employ an interleaved strategy. In T2, odd-numbered nodes send their $c_{\text{pre}}^{i+1}$ to their subsequent `device(i+1)`, and even-numbered nodes send their $c_{\text{post}}^{i-1}$ to `device(i-1)`. In T3, this pattern reverses. This approach prevents bottlenecks, optimizes channel usage, and minimizes deadlock risks. More details can be found in the pseudocode in Appendix A.2.

**Putting each module in parallel.** We tailored certain types of temporal modules to integrate into Clip parallelism, enabling distributed processing across multiple devices with efficient communication, ensuring results consistent with the original non-distributed approach:

- **Convolution module.** The temporal convolution module `Conv()` applies convolution along the temporal dimension to its input $v_{\text{in}}^{i} \in \mathbb{R}^{F_{\text{clip}} \times C'}$. In Clip parallelism, the context $c^i$ of the `Conv()` includes $c_{\text{pre}}^{i}$ and $c_{\text{post}}^{i}$. They are padded to the original sequences. Specifically, $c_{\text{pre}}^{i}$ consists of the last $n$ frames of $v_{\text{in}}^{i-1}$, and $c_{\text{post}}^{i}$ consists of the first $n$ frames of $v_{\text{in}}^{i+1}$, where $n$ is the receptive field size of the convolution.

---

[1] Only one device can communicate with another at a time.

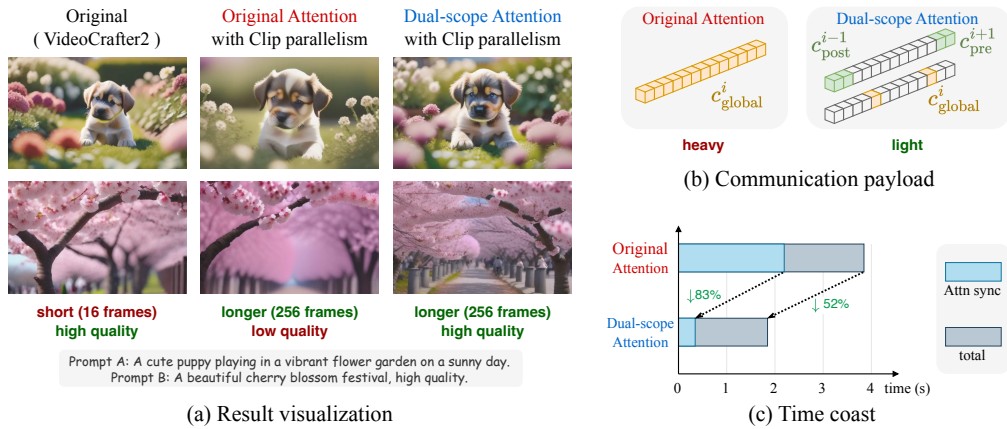

Figure 4: Comparison of video quality (a), communication payload (b), and time cost (c) between the original method and our approach with Dual-scope attention. (Videos can be viewed in the supplementary material)

- **Group normalization.** In the video diffusion model, group normalization is applied to the input tensor $v_{\text{in}}^i \in \mathbb{R}^{F_{\text{clip}} \times H \times W \times C'}$ to maintain consistent feature scaling across different frames. In Clip parallelism, each device first computes the group mean $\mu^i$ of its respective video clip. These means are aggregated to compute the global mean $\bar{\mu} = \frac{\sum_{i=1}^{N} \mu^i}{N}$, where N is the number of devices. Subsequently, using $\bar{\mu}$, each device computes its standard deviation $\bar{\sigma}^i$, which is shared to calculate the global standard deviation $\bar{\sigma}$. The global mean $\bar{\mu}$ and global standard deviation $\bar{\sigma}$, serving as $c_{\text{global}}$, are used for normalization [2].

- **Attention module.** In temporal attention modules, full self-attention requires each frame's $K$ and $V$ to be accessible on all devices. Thus, $c_{\text{global}}$ includes the K-V pairs for all frames.

## 4.2 DUAL-SCOPE ATTENTION

Applying attention in parallel inference incurs new challenges. The original attention module requires simultaneous access to all input tokens (Shaw et al., 2018). Adopting it under Clip parallelism, necessitates aggregating tokens across devices, resulting in tremendous communication costs. Additionally, as observed in Figure 4(a), attention mechanisms trained on shorter video clips often degrade in quality when extended to longer sequences.

To address these issues, we introduce the *Dual-scope attention* module. It revises the computation of K-V pairs to incorporate both local and global contexts into the attention. For each query token from frame $a$, its corresponding keys and values are computed from tokens in the frame set $\mathcal{A}^a = \mathcal{N}^a \cup \mathcal{G}$:

- *Local Context ($\mathcal{N}^a$).* This includes the $|\mathcal{N}^a|$ neighboring frames of $a$, from which the keys and values are derived to capture the local context. This local setup is typically achieved through window attention, focusing on the nearby frames to enhance the temporal coherence.

- *Global Context ($\mathcal{G}$).* In contrast, the global context consists of frames uniformly sampled from videos across all devices. This context provides keys and values from a broader range, giving the model access to long-range information.

In practice, the keys $K$ and values $V$ are constructed by concatenating the tokens from both contexts $K = \text{Concat}(K_{\text{local}}, K_{\text{global}})$ and $V = \text{Concat}(V_{\text{local}}, V_{\text{global}})$, where $K_{\text{local}}$ and $Q_{\text{local}}$ is derived from $\mathcal{N}^a$ and $K_{\text{global}}$ and $Q_{\text{global}}$ from $\mathcal{G}$. We find that this modified key-value computation can be easily incorporated into existing temporal attention without additional training, as shown in Figure 4(a), enhancing the coherence of long videos.

---

[2]Note that simply averaging the individual standard deviations $\sigma^i$ does not yield the true global standard deviation $\bar{\sigma}$.

In the implementation of Clip parallelism, the reformulated attention significantly reduces communication overhead. Instead of gathering all tokens of length $F$, we only synchronize a constant number of tokens. Specifically, we set $|c_{\text{pre}}^i| = |c_{\text{post}}^i| = \frac{|\mathcal{N}^a|}{2} = 12$ and $|c_{\text{global}}| = |\mathcal{G}| = 8$. This significantly reduces data synchronization demands while still capturing essential local and global information.

As shown in Figure 4, the results using Dual-scope attentionexhibit better video quality, and the smaller context reduces the communication payload compared to the original attention mechanism. Consequently, as illustrated in the figure, the overall inference time is reduced by 52%.

# 5 EXPERIMENTS

## 5.1 SETUPS

**Base model.** In the experiments, the text-to-video model VideoCrafter2 (Chen et al., 2024) (320 x 512) is selected as the base model of our method. VideoCrafter2, which was trained on 16-frame videos, excels at generating video clips that are both consistent and of high quality. It is also the highest-scoring open-source video generation model under the VBench (Huang et al., 2023) evaluation, achieving the top total score.

**Metrics evaluation.** VBench (Huang et al., 2023) is utilized as a comprehensive video evaluation tool, featuring a broad array of metrics across various video dimensions. For each method, videos are generated using the prompts provided by VBench for evaluation. The metrics measured encompass all the indicators under the Video Quality category in VBench, including subject consistency, background consistency, temporal flickering, motion smoothness, dynamic degree, aesthetic quality, and imaging quality. Given that VBench's evaluation is typically performed on video clips of 16 frames, we have modified the evaluation method for videos longer than 16 frames: we randomly sample five 16-frame clips from each video to evaluate separately, and then calculate the average score of these assessments.

**Baselines.** Our approach is benchmarked against several other methods:

- **FreeNoise** (Qiu et al., 2023): We chose FreeNoise as a baseline because it is also a training-free method that can base the VideoCrafter2 (Chen et al., 2024) model, which also serves as our base model, to generate long videos. It employs a rescheduling technique for the initialization noise and incorporates Window-based Attention Fusion to generate longer videos.

- **Streaming T2V** (Henschel et al., 2024): To assess our method's effectiveness in generating longer videos, StreamingT2V was chosen as our baseline. Streaming T2V involves training a new model that uses an auto-regressive approach to produce long-form videos. Like our approach, it also has the capability to generate videos exceeding 1000 frames.

   **OpenSora V1.1** (hpcaitech, 2024), a video diffusion model based on DiT (Peebles & Xie, 2023), supports up to 120 frames, can generate videos at various resolutions, and has been specifically trained on longer video sequences to enhance its extended video generation capabilities.

**Dual-scope attention setting.** In the implementation of the Dual-scope attention, the number of neighboring frames $\mathcal{N}^i$ is set to 24, with 12 frames coming from the preceding clip and 12 frames from the subsequent clip. The number of global frames, $\mathcal{G}$, is set to 8. To balance consistency and dynamics during the denoising process, the weights of frames in $\mathcal{G}$ and $\mathcal{N}^i$ are dynamically adjusted. Specifically, the weight of $\mathcal{G}$ increases by 10 for timesteps $t$ greater than 800, whereas the weight of $\mathcal{N}^i$ increases by 10 for timesteps $t$ less than or equal to 800.

**Implementation details.** By default, all parameters of the diffusion are kept consistent with the original inference settings of VideoCrafter2 (Chen et al., 2024), with the number of denoising steps set to 30. Our experiments are conducted on 8 × Nvidia 6000 Ada (with 48G memory) . To implement the temporal module in Clip parallelism, we utilized the `torch.distributed` tool package, employing Nvidia's NCCL as the backend to facilitate efficient inter-GPU communication. Additionally, all fps conditions are set to 24, and the resolution is set to $512 \times 320$. Note that the resolution for Streaming T2V cannot be modified; thus, videos are generated at its default resolution ($256 \times 256$ for preview videos and $720 \times 720$ for final videos).

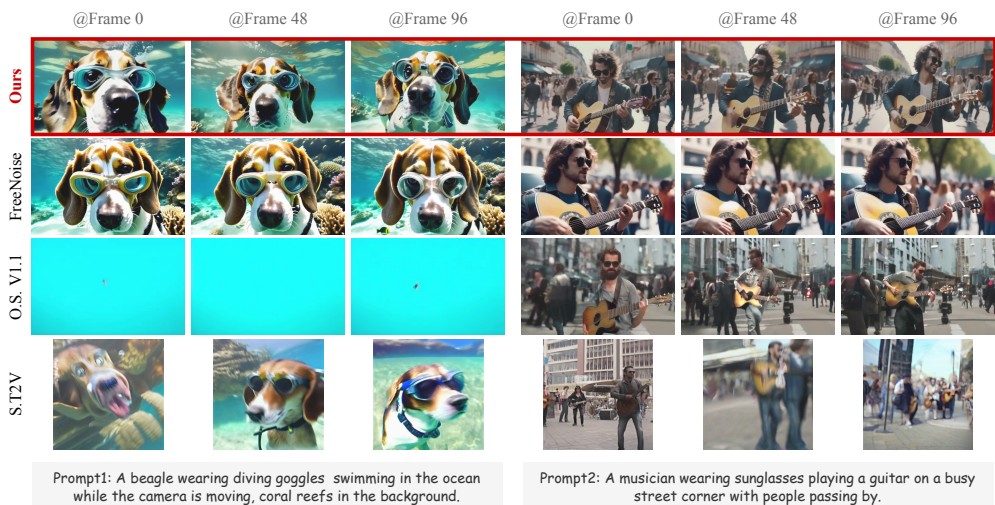

|  | @Frame 0 | @Frame 48 | @Frame 96 | @Frame 0 | @Frame 48 | @Frame 96 |

Prompt1: A beagle wearing diving goggles swimming in the ocean while the camera is moving, coral reefs in the background.

Prompt2: A musician wearing sunglasses playing a guitar on a busy street corner with people passing by.

Figure 5: Comparison of frame images from sample videos generated by different methods.

## 5.2 MAIN RESULTS

**Capacity and efficiency.**

We evaluated the capabilities of our method on an $8 \times$ Nvidia 6000 Ada (48G) setup. Our approach successfully generated videos of **2300 frames** at a resolution of $512 \times 320$, equivalent to a duration of 95 seconds at 24 frames per second. Remarkably, the entire computation process took approximately **5 minutes** (312s), benefiting from efficient communication and the leveraging of multi-GPU parallel processing.

| Method | GPU required | FPS | Time Coast (second) 128 frames | 1024 frames |
|---|---|---|---|---|
| ST2V (preview) | 1 | 0.47 | 277 | 2,196 |
| ST2V (final cut) | 1 | 0.075 | 1730 | 13,726 |
| FreeNoise | 1 | 0.64 | 201 | $\times$ |
| Open-Sora v1.1 | 1 | 0.54 | 234 | $\times$ |
| FIFO-Diffusion | 8 | 0.56 | 232 | 1,835 |
| Video-Infinity | 8 | 7.8 | 21 | 131 |

Table 1: Comparison of efficiency

Table 1 presents the capacities for long video generation of various methods, all measured under the same device specifications. To ensure comparability, we standardized the resolution of the videos generated by all methods to 512x320. For StreamingT2V, we provide two sets of data: one for generating preview videos at 256x256 resolution, and another for final videos produced at a resolution of 720x720. The results demonstrate that our method is not only able to generate ultra-long videos but also achieves unmatched speed. Furthermore, compared to the concurrent work FIFO-Diffusion (Kim et al., 2024), our method achieves more than 10 times the speed on the same 8-GPU setup.

**Video quality.** We compared the videos generated by our method with those produced by FreeNoise (Qiu et al., 2023) and StreamingT2V (Henschel et al., 2024) for long video generation. Figure 5 visualizes some frames from videos generated by different methods using the same prompt. Additionally, Table 2 displays the quality of the videos produced by these methods, evaluated across various metrics in VBench (Huang et al., 2023). More videos can be found in 1) the supplementary material, and 2) the anonymous link provided in the Appendix B.

Figure 5 shows that while the StreamingT2V (Henschel et al., 2024) method generates long videos with sufficient dynamism, they lack consistency between the beginning and end. Conversely, videos generated by FreeNoise (Qiu et al., 2023) maintain consistency in object placement throughout but exhibit minimal variation in visuals. For example, as shown in Figure 5, the video of the person playing the guitar maintains a single pose with only minimal movement. Similarly, the dog on the left remains intently focused on the camera, with no changes in the position of its ears, nose, or body. OpenSora V1.1 (hpcaitech, 2024) failed to generate the first video and the second video's background was not smooth. In contrast, our method not only ensures better consistency but also features more significant motion in the generated videos.

Table 2 reveals that our method, when compared to our base model VideoCrafter 2 (Chen et al., 2024), experiences a slight decrease in most metrics except for the metric of dynamic. In the generation of

| Method | base (VideoCrafter2) | OpenSora | FreeNoise | Video-Infinity *with C.P.* | Video-Infinity *with C.P. & D.A.* | ST2V | Video-Infinity *with C.P.* | Video-Infinity *with C.P. & D.A.* |
|---|---|---|---|---|---|---|---|---|
| Video Length | 16 frames | 64 frames | | | | 192 frames | | |
| Subject consistency | 96.85 | 86.18 | 94.16 | 90.97 | 92.74 | 75.02 | 92.34 | 90.67 |
| Background consistency | 98.22 | 95.83 | 96.63 | 94.61 | 93.90 | 87.93 | 94.78 | 92.63 |
| Temporal flickering | 98.41 | 98.47 | 98.37 | 96.38 | 97.77 | 95.96 | 94.01 | 97.10 |
| Motion smoothness | 97.73 | 97.27 | 97.04 | 95.59 | 96.84 | 94.71 | 93.21 | 95.83 |
| Dynamic degree | 42.50 | 73.61 | 44.44 | 79.17 | 81.94 | 80.56 | 88.89 | 87.50 |
| Aesthetic quality | 63.13 | 51.69 | 60.53 | 58.60 | 60.71 | 48.08 | 56.01 | 60.76 |
| Imaging quality | 67.22 | 50.61 | 67.44 | 64.00 | 67.90 | 57.85 | 64.47 | 67.79 |
| Overall consistency | 28.23 | 1.36 | 28.43 | 24.67 | 27.58 | 4.49 | 22.68 | 27.08 |
| Appearance style | 25.13 | 21.09 | 25.29 | 24.05 | 25.52 | 20.61 | 23.51 | 25.51 |
| Temporal style | 25.84 | 21.58 | 25.59 | 20.98 | 23.83 | 23.39 | 18.66 | 24.28 |
| Overall Score | 64.33 | 59.77 | 63.79 | 64.90 | 66.87 | 58.86 | 64.86 | 66.92 |

Table 2: Comparison of video quality score as benchmarked by VBench.

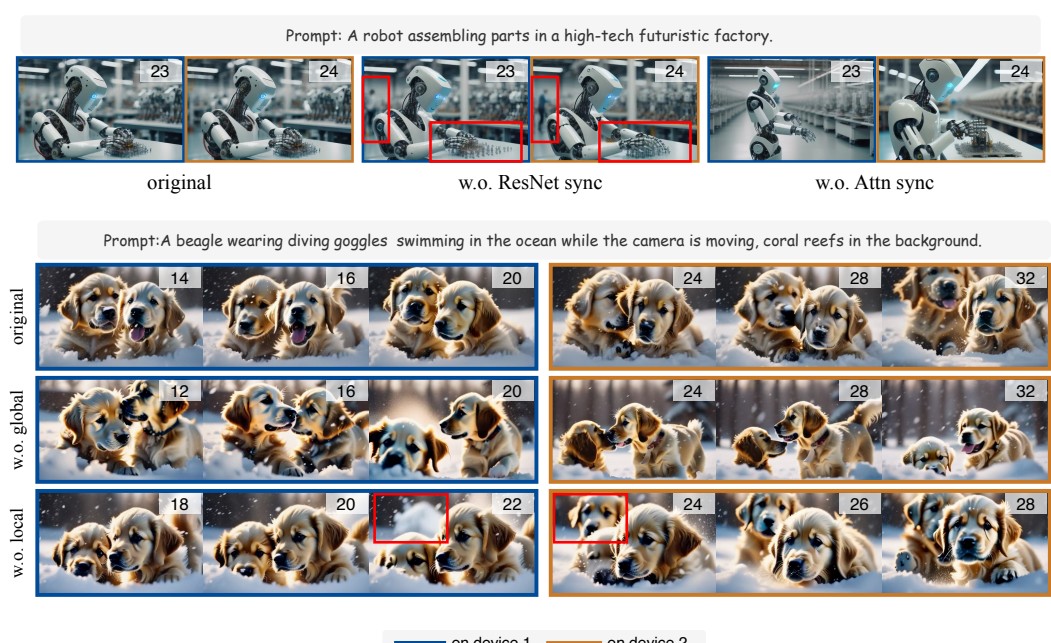

Figure 6: Visualization of ablation studies on temporal module communication and context effects in video generation. Top panel: Ablation of communication between the ResLayer module and the Attention module, showcasing two adjacent frames from the video sequence generated on different GPUs. Bottom panel: Effects of ablating different contexts within the Attention module, displaying frames from videos generated post-ablation.

64-frame videos, the performance of our method shows mixed results compared to other methods, with both advantages and disadvantages noted. However, our average metric scores are higher than those of both FreeNoise and OpenSora V1.1. In the generation of longer 192-frame videos, our method outperforms StreamingT2V across all evaluated metrics.

## 5.3 ABLATION

**Synchronization for different modules.** We performed an ablation study on the communication between the temporal `DualscopeAttn()` and `ResNet()` modules in the video diffusion model, where `ResNet()` includes temporal `Conv()` and `GroupNorm()` submodules. The top panel of Figure 6 shows that without synchronized information from `ResNet()`, discrepancies arise between frame 23 on `device(1)` and frame 24 on `device(2)`, such as differences in the color of clothes and the shape of parts held by the robot. Additionally, when synchronization is absent in `DualscopeAttn()`, frame 23 and frame 24 show a significant discontinuity. These findings

highlight the importance of synchronization in all these modules to maintain visual coherence across devices.

**Different context in Dual-scope attention.** The bottom panel of Figure 6 shows that without global context synchronization in Dual-scope attention, it becomes challenging to maintain consistent content throughout the video. For example, in frames 12 and 16 of row 2 in the figure, the ground horizon in the background remains high, but in frames beyond 20, there is a noticeable rise in the horizon, along with a lack of continuity between the video clips. Furthermore, when the local context synchronization in Dual-scope attentionis removed, although the content across different device clips remains consistent, the lack of shared context in the transition areas leads to anomalies. For instance, the content of snow in frame 22 abruptly transitions to a dog, highlighted in red. These examples highlight the importance of global and local context synchronization for video generation.

## 6 CONCLUSION

We presented Video-Infinity, a distributed inference pipeline that leverages multiple GPUs for long-form video generation. We present two mechanisms, Clip parallelism and *Dual-scope attention*, to address key challenges associated with distributed video generation. Clip parallelism reduces communication overhead by optimizing the exchange of context information, while *Dual-scope attention* modified self-attention to ensure coherence across devices. Together, these innovations enable the rapid generation of videos up to 2,300 frames long, vastly improving generation speeds compared to existing methods. This approach not only extends the practical utility of diffusion models for video production but also sets a new benchmark for efficiency in long-form video generation.

## 7 LIMITATION

To fully harness the potential of our method, it relies on the availability of multiple GPUs. Additionally, our approach does not effectively handle video generation involving scene transitions. Furthermore, our current method generates longer videos using only a single prompt or simple multi-prompt setups, which could result in less diverse content in the final video.

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

# A APPENDIX

## A.1 COMMUNICATION OVERHEAD

Table 3 demonstrates the additional time overhead caused by communication between different temporal modules. The experiments were conducted on multiple Nvidia A5000 GPUs, with two settings: a dual-GPU configuration and an eight-GPU configuration.

| Sync | Inference Time (s) | |
|---|---|---|
| | $2\times$**GPU** | $8\times$ **GPU** |
| Plain | 145.4 | 149.5 |
| + `Conv()` | 152.9 (5.1% ↑) | 157.1 (5.1% ↑) |
| + `GroupNorm()` | 158.3 (8.9% ↑) | 160.1 (7.1% ↑) |
| + `DualscopeAttn()` | 170.7 (17.4% ↑) | 180.2 (20.5% ↑) |
| Full Sync | 182.3 (25.3% ↑) | 192.3 (28.6% ↑) |

Table 3: Effect of Synchronization on Inference Time

## A.2 COMMUNICATION ALGORITHM

---
**Algorithm 1** Distributed Temporal Module Communication

---
**Require:** $i$ (the ID of the device), $v_{\text{in}}^i$ (the input latent segment)
**Ensure:** Seamless and efficient distribution of frames for video processing.
1: Prepare the global context $c_{\text{global}}^i$ using $v_{\text{in}}^i$
2: `dist.all_gather(`$c_{\text{global}}^i$`)`
3: **if** $i$ mod 2 == 1 **then**
4:     $c_{\text{pre}}^i =$ `dist.recv(i+1)`
5:     Prepare the local context for `device(i+1)` using $v_{\text{in}}^i$
6:     `dist.send(`$c_{\text{post}}^{i+1}$`)`
7:     $c_{\text{post}}^i =$ `dist.recv(i-1)`
8:     Prepare the local context for `device(i-1)` using $v_{\text{in}}^i$
9:     `dist.send(`$c_{\text{pre}}^{i-1}$`)`
10: **else**
11:     $c_{\text{post}}^i =$ `dist.recv(i-1)`
12:     Prepare the local context for `device(i-1)` using $v_{\text{in}}^i$
13:     `dist.send(`$c_{\text{pre}}^{i-1}$`)`
14:     $c_{\text{pre}}^i =$ `dist.recv(i+1)`
15:     Prepare the local context for `device(i+1)` using $v_{\text{in}}^i$
16:     `dist.send(`$c_{\text{post}}^{i+1}$`)`
17: **end if**

---

# B GALLERY

More videos are available in the supplementary materials and at the following link: Anonymous Link

