# OpenReview forum: "Video-Infinity: Distributed Long Video Generation"
_ICLR.cc/2025/Conference — ICLR 2025 Conference Withdrawn Submission_

### Official Review · Reviewer_4vVA · 2024-11-02

**Soundness:** 3
**Presentation:** 3
**Contribution:** 3
**Rating:** 6
**Confidence:** 4

**Summary:**

The paper presents Video-Infinity, a novel framework for generating long-form videos using distributed diffusion models across multiple GPUs. This approach aims to reduces the inference time and resource demands typically associated with long video generation. The paper proposed two methods: Clip parallelism and Dual-scope attention, which optimize inter-GPU communication and temporal attention across frames, respectively. The methodology enables the generation of videos up to 2,300 frames in just 5 minutes,  faster than existing methods.

**Strengths:**

1. The integration of Clip parallelism and Dual-scope attention is a novel approach that effectively addresses the scalability and efficiency challenges in video generation.

2. The paper demonstrated ability to generate longer videos much faster than current methods, achieving substantial reductions in generation time.

3. Experiments are conducted to validate the performance, showcasing significant improvements over other methods in terms of speed and video length capabilities.

**Weaknesses:**

1. The method of synchronizing context across GPUs, crucial for maintaining temporal coherence, is not discussed detail.

2. While the framework improves efficiency, there is not much discussion on how these gains impact the qualitative aspects of the videos, such as resolution, realism, particularly under complex scene dynamics.

**Questions:**

1. How the synchronization latency affects the continuity and quality of the video, especially in dynamic scenes? Are there mechanisms in place to mitigate negative impacts if synchronization is delayed?

2. How does the proposed model perform with high-motion sequences or videos requiring rapid scene changes, and what are the limitations of the current approach in handling such dynamics?

3. The paper focuses on generating videos from a limited set of prompts and scenarios. How well does the approach generalize to a wider variety of video content or more complex scenes?

---

### Official Review · Reviewer_x3cC · 2024-11-02

**Soundness:** 2
**Presentation:** 2
**Contribution:** 1
**Rating:** 3
**Confidence:** 4

**Summary:**

The paper introduces Video-Infinity, a distributed inference framework designed to enable efficient generation of long videos using diffusion models. It tries to address a challenge in video generation: the resource-intensive nature of long-form content, which often restricts video length and quality due to memory and computation limits. The proposed approach leverages two main innovations: Clip parallelism, which optimizes the distribution and synchronization of context information across multiple GPUs, and Dual-scope attention, which balances local and global temporal self-attention to maintain semantic coherence without requiring additional model training.

**Strengths:**

1. The empirical results are persuasive, with Video-Infinity achieving a 10x improvement over comparable methods like FIFO-Diffusion and being significantly faster than alternatives like Streaming T2V.

2. The paper is well-organized, clearly outlining the technical details, methodology, and communication strategies.

**Weaknesses:**

1. It looks like this work adopt the idea from DistriFusion [1]. While the authors claim to tackle a more challenging problem, the dimensionality of frames, from a technical standpoint, is actually much simpler to manage compared to the problems addressed in DistriFusion.

2. How does this method impact frame-to-frame continuity? I noticed that many of the generated videos in the Supplementary Material exhibit noticeable continuity issues. The authors do not seem to have adequately addressed this problem. Additionally, many other generated long videos can only display repetitive motions and clips.

3. The evaluation lacks comprehensiveness, as the authors have only demonstrated their method on a single model, VideoCrafter2. It remains unclear whether the approach is effective across a broader range of model architectures. For instance, how well does this method generalize to new architectures like DiT? Additionally, what is the performance impact on these models?

4. It's more of an engineering work, the novelty contribution of this work is not good enough.

I'm sure it needs dedicated effort for applying this method on every new model architecture.

[1] DistriFusion: Distributed Parallel Inference for High-Resolution Diffusion Models, CVPR'24

**Questions:**

Please refer to Weaknesses.

---

### Official Review · Reviewer_tAHD · 2024-11-03

**Soundness:** 3
**Presentation:** 3
**Contribution:** 2
**Rating:** 3
**Confidence:** 5

**Summary:**

This paper presents Video-Infinity, a distributed inference pipeline for long-video generation using diffusion models. It introduces two techniques for the main challenges in long-video generation. Clip parallelism divides a long clip generation task into several short clips to address the high GPU memory usage. Dual-scope attention gathers local and global context for temporal self-attention to generate a consistent long video. It compares with FreeNoise, StreamingT2V, and OpenSora 1.1V.

**Strengths:**

1. The paper is well-written and easy to follow.
1. It is a training-free inference pipeline while extending the baseline model generation capacity.
2. ***Dual-scope Attention*** provides a new view of gathering the global and local context for high-fidelity long video generation. The generation results are impressive. It might provide insight into the training scheme or new architecture design.

**Weaknesses:**

1. The novelty of **Clip Parallelism** is limited. The paper merely migrates the DistriFusion[1] to the video diffusion model, where DistriFusion splits a large image into patches while this paper splits a long video into short clips. The distributed modules are similar to the sparse operations in DistriFusion[1], except for extending the sparse 2D convolution to the 1D/3D temporal convolution with different padding schemes. Also, the *GroupNorm* modification is similar. Moreover, the DistriFusion[1] further introduced *Corrected asynchronous GroupNorm*, which is more efficient than the paper's implementation since the asynchronous communication can be pipelined into the computation.
1. The paper didn't compare the video quality with FIFO-Diffusion[2], which also focused on long-video generation. It is difficult to demonstrate the proposed method's advantage over the SOTA work.
1. In the comparison of efficiency, comparing Open-Sora v1.1 and the proposed method is unfair because they use different model architectures (Spatial-Temporal DiT vs. VideoCrafterV2).
1. There are several typos. (e.g., GPU index in Fig.1, captions in Fig.3

[1] DistriFusion: Distributed Parallel Inference for High-Resolution Diffusion Models. Muyang Li, Tianle Cai, et al. CVPR 2024

[2] FIFO-Diffusion: Generating Infinite Videos from Text without Training. Jihwan Kim and Junoh Kang and Jinyoung Choi and Bohyung Han, NeurIPS 2024

**Questions:**

1. The paper focuses on the video diffusion model with the temporal self-attention layer. However, DiT-based architectures are popular in current work, and they tend to use the 3D attention layer instead of the temporal self-attention layer (e.g., CogVideo-5B[2]).
How does the *Dual-scope Attention* perform on the 3D attention layer?
2. How is the video quality compared with FIFO-Diffusion[2]?
3. Would it be possible to extend Video-Infinity to OpenSora for the efficiency comparison?


[3] CogVideoX: Text-to-Video Diffusion Models with An Expert Transformer. Zhuoyi Yang, Jiayan Teng, et al, arxiv 2408.06072. 2024.

---

### Official Review · Reviewer_pci8 · 2024-11-04

**Soundness:** 3
**Presentation:** 3
**Contribution:** 2
**Rating:** 5
**Confidence:** 3

**Summary:**

The paper proposes Video-Infinity, a distributed inference pipeline designed for long-form video generation using diffusion models. The framework leverages two main mechanisms: Clip parallelism, which distributes video segments across multiple GPUs to improve processing efficiency, and Dual-scope attention, which balances local and global temporal contexts across devices. Together, these components enable Video-Infinity to generate lengthy, coherent videos with reduced memory overhead. On an 8 × Nvidia 6000 Ada GPU setup, the framework can produce videos up to 2,300 frames in approximately 5 minutes.

**Strengths:**

1. This work brings incremental novelty by adapting distributed parallelism specifically for long-form video generation. It introduces a dual-scope attention mechanism to balance local and global temporal interactions, ensuring coherence across extended sequences. The clip parallelism approach further enables efficient processing of video clips across GPUs, effectively handling the unique scalability and memory demands of video data. These adaptations, including optimizations for temporal continuity, showcase Video-Infinity’s tailored application of distributed inference to the distinct challenges of generating coherent long videos.

2. Speed up performance is great. The proposed Clip parallelism and Dual-scope attention mechanisms optimize inter-device communication and memory requirements, leading to faster processing times and scalability for generating extended video sequences. It could reduce the inference time by up to 52%.

**Weaknesses:**

1. Performance. In the Table 2 under 64 frames settings, although the proposed work got the highest overall score, it did not showed dominating better results than other baselines.

2. Results on longer context. This work claims capability to generate longer video clips, while it only shows results for a maximum of 192 frames in Table 2. Since it emphasis the long video generation ability, I would suggest putting more quantitive results on longer video.

3. Results on memory usage comparison. This work lacks of comparison of reduced memory overhead to demonstrate the efficiency of the method.

**Questions:**

Has “Video-Infinity” been tested on different types of video content, such as fast-moving scenes or varying lighting conditions, which might challenge the coherence of frame transitions? How robust is the model across these diverse scenarios?

---

### Note · Authors · 2024-11-26

I have read and agree with the venue's withdrawal policy on behalf of myself and my co-authors.